# Organic Fertilization of Growing Media: Response of N Mineralization to Temperature and Moisture

**Patrice Cannavo** [1],*[ID], **Sylvie Recous** [2][ID], **Matthieu Valé** [3], **Sophie Bresch** [4], **Louise Paillat** [1][ID], **Mohammed Benbrahim** [5] and **René Guénon** [1]

1    Institut Agro, EPHOR, 49000 Angers, France; louisepaillat@gmail.com (L.P.); rene.guenon@agrocampus-ouest.fr (R.G.)
2    Université de Reims Champagne Ardenne, INRAE, FARE, UMR A 614, 51097 Reims, France; sylvie.recous@inrae.fr
3    AUREA AGROSCIENCES, 45160 Ardon, France; m.vale@aurea.eu
4    CDHR Centre-Val de Loire, Domaine de Cornay, 45590 Saint-Cyr-en-Val, France; sophie.bresch@astredhor.fr
5    RITTMO Agroenvironnement, ZA Biopôle, 37 rue de Herrlisheim, CS 80023, CEDEX, 68025 Colmar, France; mohammed.benbrahim@rittmo.com
*    Correspondence: patrice.cannavo@agrocampus-ouest.fr

**Abstract:** Managing plant fertilization is a major concern of greenhouse growers when it comes to sustainable production on growing media. Organic fertilization is popular, but more difficult to control since organic compounds first need to be mineralized by microbes. The objective of this study was to characterize the time course of N mineralization by different fertilizer–growing media pairs, in the absence of plants. Several incubations were carried out at four temperatures (4, 20, 28, and 40 °C) and three suction potentials (−3.2, −10, and −31.6 kPa) on four growing media under two organic fertilization conditions to study the dynamics of $NH_4^+$ and $NO_3^-$ production. The results showed that the release of mineral N was strongly dependent on growing media, temperature, humidity, and fertilizer nature, varying from 10.7% to 71.3% of the N fertilizer applied. A temperature action law was established for the four growing media. The Q10 value of the growing media was 1.13, lower than the average Q10 value of arable soils. On the other hand, the specific behavior of the growing media did not yield a single humidity action law. Nevertheless, the nitrification process, evaluated by analyzing the ratio of $NO_3^-$ to total mineral N, showed a humidity-dependent relationship common to the four growing media and comparable to admitted observations on soils. Nitrification was optimal when growing media humidity was higher than 0.46 *v/v*.

**Keywords:** $NH_4^+$; $NO_3^-$; nitrification; Q10; modeling

## 1. Introduction

Consumers are concerned about food quality and the environmental impact of its production. The subject is thorny in horticulture, particularly in soilless production which consumes resources (water and other inputs). As a consequence, producers are moving toward agro-ecological practices such as organic fertilization and the development of growing media from renewable organic materials [1]. Indeed, organic fertilization introduces a recycling concept in agroecosystems, and the non-use of synthetic inorganic N fertilizers considerably reduces the $CO_2$ emissions produced during the industrial $N_2$ fixation.

In conventional soilless production (cultivation in pots and containers), the plant grows in a finite volume of a growing medium with limited buffering capacity for water, temperature, and pH in particular [2]. The physical, chemical, and, to a lesser extent, biological properties of growing media materials have been investigated over the last 40 years, but practical considerations have been relatively little investigated [3]. Professionals have good knowledge of the physicochemical properties of the growing media, allowing for the control of irrigation and mineral fertilization. Introducing organic fertilizers requires

adapted practices because organic fertilizers first have to be mineralized by the microbiota of the growing media before being assimilated by plants.

Matching the rate of nutrient release by micro-organisms to the plant demands is essential [4,5]. Although microbial communities are widely used in growing media, few studies have characterized them. The authors of [6,7] studied microbial communities in peat, coir, and wood fiber growing media. They showed that organic growing media display specific activities and microbial structures depending on their origin and manufacturing process. Organic nutrient sources may be single or blended, and they may come from plant or animal byproducts or allowable mined sources [5]. Solid organic fertilizers are often unbalanced in their nutrient content, especially in their nitrogen, phosphorus, and potassium ratios, and a delay in shoot growth can result from their use [8,9]. The authors of [10] studied microbial activities involved in organic C, N, P, and S availability and the release of mineral forms in different growing media made of organic fertilizer combinations. Specific responses were observed, showing the complexity of the mineralization process. In particular, the mineralization rate varied greatly from one growing media to another.

Mineral nitrogen is the nutrient most used by plants, and it is often the most limiting element for plant growth [11]. The N preference of plants is variable and closely related to environmental conditions. For example, the N preference of plants changes from $NO_3^-$ to $NH_4^+$ from drier to wetter sites [12], while the preference shifts from $NH_4^+$ under an acidic environment to $NO_3^-$ at alkaline locations [13]. Moreover, $NO_3^-$ is more accessible to plants because it reaches the roots by mass flow, whereas ammonium reaches them by diffusion [14]. Nitrogen deficiency results in symptoms such as reduced growth, and yellowing of leaves occurs very fast after the onset of deficiency [15]. Conversely, nutrient excess due to too rapid mineralization of organic fertilizer or the presence of unwanted ions such as $SO_4^{2-}$, sodium, or chlorine at high concentrations can result in salinization of the organic growing media [16]. It is difficult for growers to match the availability of dissolved growing media nutrients with plant demands at different stages of their developmental cycle [17]. Because a large proportion of organic nutrients are mineralized within the first few weeks and can leach out of containers, substrates containing organic fertilizers are typically used as the sole fertilizer source only for short-term crops. For long-term crops, substrates containing fertilizers are typically not enough to supply plant needs throughout the crop cycle and must be supplemented by top-dressing, side-dressing, or the use of liquid fertilizers in irrigation water [1].

The microbe-mediated N mineralization rate and the microbial community composition are highly variable and dependent on several factors such as growing media temperature, air porosity, and moisture content, as well as on the nature of the organic fertilizer source and the growing media composition (particle size and composition) [17–19]. The C:N ratio of growing media can also impact organic fertilizer mineralization. Substrates with C:N ratios exceeding 30:1 tend to immobilize N due to microbial decomposition of available C, which requires N [20]. Wood components such as composted barks, hammer milled, wood materials, and sawdust can have C:N ratios of 300:1 or more, and they have a high potential to immobilize N from applied fertilizers. Non-wood components with high C:N ratios, such as coir fiber, can also immobilize N [21].

N mineralization and nitrification have been thoroughly studied in soils, but knowledge gaps persist with regard to growing media; as they present low biodegradability, one can wonder whether indigenous microbial communities are suitable for organic N mineralization and nitrification, depending on temperature and moisture conditions. More generally, the transposition of mineralization and nitrification knowledge from soils to growing media is questioned. Thus, understanding the drivers of organic fertilizer mineralization and nitrification in horticultural growing media is necessary for a better prediction of mineral N availability for plants. The objectives of this study were to characterize the dynamics of $NH_4^+$ and $NO_3^-$ production, and to evaluate the impacts of temperature and humidity on the N dynamics in the growing media. The ambition was to set up temperature and moisture functions to be ultimately used for modeling the N dynamics

in fertilized growing media under fluctuating conditions of moisture and temperature, as met by producers. A laboratory incubation experiment was conducted to characterize the organic N fertilizer mineralization of four commercial growing media under different growing media moisture and temperature regimes. We hypothesized that the action laws for temperature and moisture established for soils would be applicable to the growing media, whatever the growing media.

## 2. Materials and Methods

### 2.1. Materials

Four marketed growing media (GM1 to GM4) were studied. They were selected to be representative of the mostly used growing media (frequency and commercial volume, growing media producers survey, results not shown). Their properties are presented in Table 1. GM1 was made of black peat and composted plant fibers (80–20 vol.%), GM2 contained blond and black peat, coconut fiber, and composted plant fibers (50–20–10–20 vol.%), GM3 contained blond peat, coconut fiber, and composted bark (70–20–10 vol.%), and GM4 contained blond and black peat, coconut fiber, and green waste compost (60–10–20–10 vol.%). The properties of GM3 were somewhat different from those of the other growing media, with a coarser particle size, a higher OM content, a higher C:N ratio, and a lower bulk density. Three fertilizer modalities were studied: no fertilization (F0), organic fertilizer of an animal-based origin (F1), and organic fertilizer of a plant-based origin (F2). The fertilizer compositions are presented in Table 2, but commercial names were kept confidential.

**Table 1.** Physicochemical properties of the growing media.

|  | GM1 | GM2 | GM3 | GM4 |
|---|---|---|---|---|
| Professional Use | Container/Aromatic and Flowering Plants | Market Garden Plants in Plugs or Trays | Container/Tree and Shrub | Market Garden Plants in Plugs or Trays |
| Particle size distribution |  |  |  |  |
| Size fraction >4 mm (%) | 6.1 | 4.7 | 43.3 | 5.4 |
| Size fraction 2–4 mm (%) | 10.1 | 12.7 | 11.6 | 9.3 |
| Size fraction <2 mm (%) | 83.8 | 82.6 | 45.1 | 85.3 |
| Bulk density (g·cm$^{-3}$) [1] | 0.18 | 0.20 | 0.12 | 0.18 |
| Total porosity (*v/v*) [2] | 89.1 | 88.3 | 92.0 | 88.0 |
| EAW (*v/v*) [3] | 0.38 | 0.41 | 0.23 | 0.40 |
| AFP (*v/v*) [4] | 0.17 | 0.03 | 0.47 | 0.04 |
| pH water [5] | 6.8 | 6.7 | 7.3 | 6.5 |
| EC (mS·cm$^{-1}$) [6] | 0.7 | 0.7 | 0.6 | 0.6 |
| OM (g dw·kg$^{-1}$) [7] | 687.3 | 690.6 | 908.5 | 712.6 |
| Org N (g dw·kg$^{-1}$) | 11.1 | 11.1 | 6.9 | 13.1 |
| C:N ratio | 30.9 | 31.0 | 65.9 | 27.2 |
| Total N (g dw·kg$^{-1}$) [8] | 11.5 | 11.5 | 6.9 | 13.4 |
| AmoA (log nb_seq·g$^{-1}$) | 8.06 | 7.96 | 6.60 | 8.02 |
| Basal respiration (µg C-$CO_2$·g$^{-1}$ dw·h$^{-1}$) [9] | 0.30 ± 0.1 | 0.80 ± 0.32 | 0.85 ± 0.1 | 0.99 ± 0.2 |

[1] Bulk density (g·cm$^{-3}$) was determined following [22], and [2] total porosity (*v/v*) was determined following [22], [3] EAW: easy available water (%, *v/v*) and [4] AFP air-filled porosity (%, *v/v*) were calculated from water retention curves determined using a tension table draining at pressure potentials ranging from −1 to −10 kPa [22]; [5] pH was determined following [23]; [6] EC: electrical conductivity was determined following [24]; [7] OM (organic matter; % dry mass) was determined by loss of ignition (550 C, 7 h); [8] total N was determined by dry ignition according to [25]; [9] basal respiration was obtained by the Microresp[TM] method with growing media maintained at 60% of the water holding capacity at 25 °C for 1 week.

### 2.2. Experimental Design

The four growing media were incubated at four temperatures (4, 20, 28, and 40 °C) and three matric suctions (humidity maintained at −3.2, −10.0, and −31.6 kPa corresponding to pF1.5, pF2.0, and pF2.5, respectively), with or without added fertilizer, in the dark, in the absence of plants. Destructive samples were used to measure $NH_4^+$-N and $NO_3^-$-N contents after 3, 7, 14, 28, and 49 days. They consisted of 90 mL vials filled with growing media depending on its bulk density (Table 1), i.e., 16, 18, 11, and 17 g dw per vial for GM1, GM2, GM3, and GM4, respectively. Vials were destroyed at each date for measurements.

They were placed in trays with a lid on each vial, and the growing media water content was maintained by weighing control. The organic fertilizer was applied at a rate of 55 g N·kg$^{-1}$ growing media dw. Caps were placed on the vials, unsealed to permit air circulation but limit fast water evaporation. The amount of applied fertilizer was calculated on the basis of usual producers' practices (200 g fertilizer N·m$^{-3}$). The growing media were analyzed for their $NH_4^+$-N and $NO_3^-$-N contents at the beginning and after the start of the experiment. Three replicates were prepared per modality.

**Table 2.** Fertilizer composition.

|  | **F1** | **F2** |
|---|---|---|
| OM (g·kg$^{-1}$) | 559.5 | 636.7 |
| C (g·kg$^{-1}$) | 279.8 | 318.4 |
| Org N (g·kg$^{-1}$) | 59.5 | 54.1 |
| Total N (g·kg$^{-1}$) | 67.4 | 55.1 |
| Total P (mg·kg$^{-1}$) | 35.4 | 18.6 |
| Total K (mg·kg$^{-1}$) | 52.3 | 40.8 |
| Total Mg (mg·kg$^{-1}$) | 6.4 | 5.2 |
| Total Mn (mg·kg$^{-1}$) | 50.8 | 258.8 |
| C:N ratio | 3.8 | 5.1 |
| C:P ratio | 7.9 | 17.1 |
| N:P ratio | 1.9 | 3.0 |
| pH | 6.8 | 6.9 |

OM was determined following [26]; inorganic N was extracted with 1 M KCl (1:5 *v/v* ratio). Ammonium and nitrate concentrations were determined by colorimetry using a continuous flow analyzer (Skalar Analytical). Total P, K, Mg, and Mn were extracted following [27] and measured following [28]; pH was measured following [23].

### 2.3. Analysis

$NH_4^+$-N and $NO_3^-$-N were extracted with deionized water (1:1.5 vol.) for 1 h. Concentrations were determined by colorimetry using a continuous flow analyzer (Skalar Analytical). To obtain a growing media water suction curve, samples were saturated with distilled water for 48 h, with three replicates per growing media. Then, they were gradually dried using sand suction tables [29] with potentials equivalent to 0, −3.2, and −10 kPa. A ceramic pressure press was used for suction equivalent to −31.6 kPa [30,31]. When equilibrium was reached (2–3 days), the samples were dried in an oven at 105 °C for 48 h and weighed.

Nitrifying bacteria were quantified as follows: total nucleic acids were extracted from growing media samples using a Qiagen DNEasy PowerSoil kit (Cat No./ID: 12888-100). Then, quantitative polymerase chain reaction was performed using primers 968 R and 1401 R [32] for total bacteria and primers amoA-1F and amoA-2R [33] for nitrifying bacteria.

### 2.4. Data Treatments

#### 2.4.1. Ammonium and Nitrate Concentrations

N mineralization was estimated by measuring $NH_4^+$-N and $NO_3^-$-N concentrations, and their sum was used as the total mineral N concentration at each sampling time. Since growing media already contained mineral N (Table 1), $NH_4^+$-N and $NO_3^-$-N concentrations were expressed by subtracting each respective initial mineral N content. Nitrate was the final product of N mineralization we monitored; hence, we expressed $NO_3^-$ production as the relative proportion of total mineral N (i.e., $NO_3^-$ + $NH_4^+$) released from fertilizer degradation at each timepoint.

#### 2.4.2. Abiotic Factors

The temperature action law was determined in two steps, as described below.

First, it corresponded to a ratio of the mineralization rate at a given temperature over the mineralization rate at a reference temperature. Second, this ratio was plotted against

growing media temperature, and modeled using the STICS crop model temperature action law dedicated to the simulation of organic matter decomposition [34].

$$f(T) = \frac{\left([N]_{i,fin} - [N]_{i,init}\right)}{\left([N]_{ref,fin} - [N]_{ref,init}\right)} = \frac{B}{1 + C \times exp(-k \times T)}, \tag{1}$$

where $T$ is the temperature (°C), $[N]_{i,fin}$ and $[N]_{i,init}$ are the total mineral N on days 49 and 0 at a given temperature $T_i$, respectively, and $[N]_{ref,fin}$ and $[N]_{ref,init}$ are the total mineral N on days 49 and 0 at the reference temperature $T_{ref}$ = 20 °C, respectively. This reference temperature was made to be close to that of the STICS model (i.e., 15 °C). B was a dimensionless adjusted parameter, and k was an adjusted parameter (°C$^{-1}$). C was a parameter and was recovered by solving the following equation:

$$C = (B - 1) \times \exp\left(k \times T_{ref}\right), \quad C = (B - 1) \times \exp\left(k \times T_{ref}\right). \tag{2}$$

The water content action law was also determined in two steps, as described below.

First, it corresponded to a ratio of the mineralization rate at a given growing media matric suction over the mineralization rate at a reference matric suction. Second, this ratio was plotted against growing media moisture, and modeled using the STICS crop model moisture action law dedicated to the simulation of organic matter decomposition [34].

$$f(H) = \frac{\left([N]_{i,fin} - [N]_{i,init}\right)}{\left([N]_{ref,fin} - [N]_{ref,init}\right)} = \frac{H - H_{wp} \times H_{fc}}{\left(H_{fc} - H_{wp}\right) \times H_{fc}}, \tag{3}$$

where $H$ is the volumetric water content of the growing medium ($v/v$), $[N]_{i,fin}$ and $[N]_{i,init}$ are the total mineral N on days 49 and 0 at a given water content $H_i$, respectively, and $[N]_{ref,fin}$ and $[N]_{ref,init}$ are the total mineral N on days 49 and 0 at the reference water content corresponding to water suction −10 kPa, respectively. $H_{wp}$ is the volumetric water content at the wilting point (water suction −100 kPa), and $H_{fc}$ is the volumetric water content at field capacity (water suction −1 kPa).

To further understand the temperature and moisture interactions, the ratio of $NO_3^-$ to total mineral N was calculated as a mean ratio for the whole incubation period. This allowed us to identify abiotic conditions that may have slowed down or favored the nitrification process.

### 2.5. Statistical Analyses

We used three-way repeated-measures ANOVA (rmANOVA) to test the interaction of the growing medium type, the fertilizer type (Fert), temperature (Temp), and matric water suction ($\psi$) on $NO_3^-$, $NH_4^+$, and $NH_4^+ + NO_3^-$ concentrations following fertilization. We analyzed these effects separately depending on the significant interactions. We present the results for each growing medium, comparing temperatures or matric water suctions, and only with or without addition of fertilizers 1 or 2 to simplify the viewing of these effects. Significant differences were tested by the least significance difference test (LSD, $p < 0.05$). Correlations were tested using Pearson correlations ($p < 0.05$). When data seemed to present segmented regressions, we tested piecewise regressions with SegReg free software.

### 3. Results

### 3.1. Dynamics of Growing Media N Mineral Content

We did not find a significant four-level interaction among growing medium, fertilizer type (Fert), temperature (Temp), and humidity (Hum) over time as hypothesized (Table S1, four-way repeated-measures ANOVA). Instead, we did find significant three-way interactions such as GM × Temp × Hum. This interaction was the most powerful one (F > 3.9, $p < 0.001$, within effect) and was also confirmed independently of time

(GM × Temp × Hum: F > 1.7, $p < 0.001$, between effect). As a result, we focused on these interaction factors to present our results (i.e., without considering the fertilizer type, even though we detected some minor modularity of the results per fertilizer type compared to GM, Temp and Hum).

Temperature significantly controlled the total mineral N content (Figure 1), as well as $NH_4^+$ and $NO_3^-$ contents (Figures S1 and S2, respectively), in the four growing media. The pattern of total mineral N differed depending on the growing media. We generally observed a plateau between 28 and 49 days modulated by temperature, except GM1 and GM2 that displayed a linear increase in mineral N content at 40 °C. GM1 and GM2 presented the best mineral N content at 40 °C after 49 days of incubation (877 and 807 mg N·kg$^{-1}$ dw growing media, respectively, Tables S2 and S3). The temperature increase from 4 to 20 °C was always significant, whereas 20 °C and 28 °C tended to have similar effects on GM1 and GM2. For GM3 and GM4, 28 °C gave the best mineral N content, whereas 40 °C gave a lower content. In the absence of organic fertilization, the rates were close to zero except for GM1 and GM2, where total mineral N significantly increased at 40 °C; moreover, a negative N content was found in GM2 and GM4, especially at 28 °C, corresponding to organization of the initial mineral N content. At 28 °C, GM3 contained the highest mineral N content reached in these incubations (1053 mg N·kg$^{-1}$ dw growing media, Table S4).

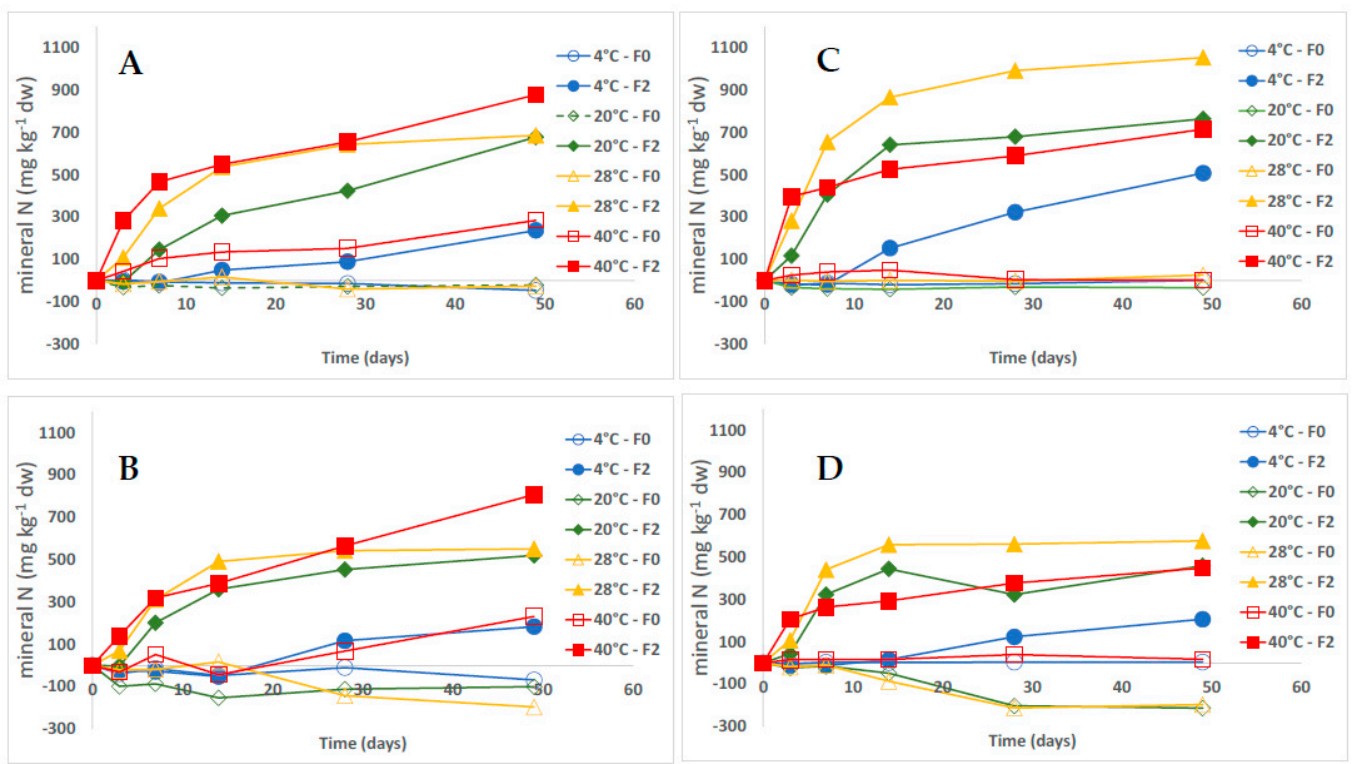

**Figure 1.** Influence of temperature on net N mineralization at −10 kPa water matric suction, for (**A**) GM1, (**B**) GM2, (**C**) GM3, and (**D**) GM4, with fertilizer F2 or without fertilizer (F0).

Humidity significantly controlled the mineral N content (Table S1, Figure 2), as well as $NH_4^+$ and $NO_3^-$ contents (Figures S3 and S4, respectively), in the four growing media, but less markedly so than temperature. We observed similar trends, with a decreasing mineral N content between 28 and 49 days of incubation. GM1 showed the best mineral N content at −3.2 kPa (the highest humidity rate) and the lowest one at −31.6 kPa (the lowest humidity rate), whereas GM2 presented its best mineral N content at −31.6 kPa. GM3 showed very contrasted mineral N dynamics from 0 to 28 days, and finally reached the same level of mineral N content after 49 days of incubation whatever the humidity level (Figure 2). GM4 showed the highest contrasts between humidity levels, with −31.6 kPa giving the highest mineral N content and −3.2 kPa giving the lowest one (712 mg N·kg$^{-1}$ dw GM

and 378 mg N·kg$^{-1}$ dw growing media, respectively, Figure 2; Table S5). In the absence of fertilization, GM1 provided mineral N at −3.2 kPa (151 mg N·kg$^{-1}$ dw growing media), while GM2 provided a similar content at −31.6 kPa, indicating that humidity did not drive the mineral N content in the same way as in GM1. In GM4, we observed a strong organization of the mineral N content, with no significant effect of humidity (Figure 2, Table S5).

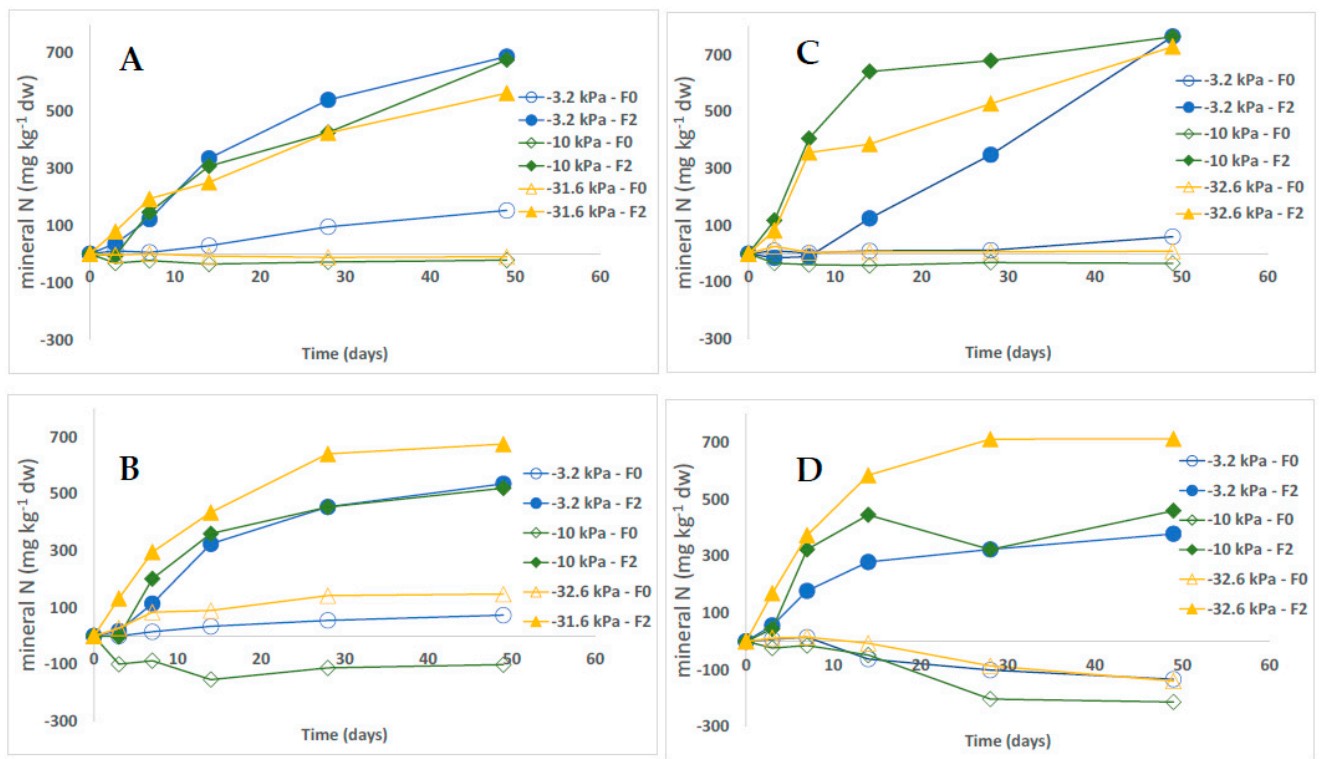

**Figure 2.** Influence of water matric suction on net N mineralization at 20 °C, for (**A**) GM1, (**B**) GM2, (**C**) GM3, and (**D**) GM4, with fertilizer F2 or without fertilizer (F0).

### 3.2. Fertilizer Mineralization

An increase in temperature from 4 °C induced an increase in the percentage of fertilizer N mineralized (Table 3), with slightly contrasting results depending on the growing medium, the fertilizer type, and humidity leading to 20, 28, or 40 °C with the highest percentage mineralization of the applied fertilizer N. Whatever the factor, F2 mineralized faster than F1 (10.7–69.2% vs. 14.7–71.3%). GM1 and GM4 reached the highest percentage of fertilizer mineralization at −31.6 kPa, but at different temperatures (40 °C and 20 °C, respectively). GM2 and GM3 reached the highest percentage of fertilizer mineralization at −10 kPa and 28 °C.

**Table 3.** Total mineralized fertilizer F1 and F2 as a percentage of applied N on day 49 (i.e., at the end of the experiment) ($n = 3$, SD = standard deviation).

| suction (kPa) | T | GM1 F1 Mean | GM1 F1 SD | GM1 F2 Mean | GM1 F2 SD | GM2 F1 Mean | GM2 F1 SD | GM2 F2 Mean | GM2 F2 SD | GM3 F1 Mean | GM3 F1 SD | GM3 F2 Mean | GM3 F2 SD | GM4 F1 Mean | GM4 F1 SD | GM4 F2 Mean | GM4 F2 SD |
|---|---|---|---|---|---|---|---|---|---|---|---|---|---|---|---|---|---|
| −3.2 | 4 | 10.7 | 1.2 | 14.7 | 1.3 | 31.7 | 2.7 | 36.5 | 3.3 | 20.3 | 1.4 | 35.0 | 0.5 | 14.5 | 0.8 | 35.0 | 8.2 |
| | 20 | 28.2 | 0.9 | 43.1 | 3.7 | 32.3 | 1.6 | 42.5 | 0.6 | 29.2 | 2.7 | 37.5 | 0.5 | 26.1 | 1.4 | 43.0 | 2.5 |
| | 28 | 25.0 | 0.3 | 42.7 | 0.7 | 24.2 | 1.1 | 39.9 | 0.6 | 32.7 | 1.4 | 40.0 | 1.3 | 39.7 | 0.8 | 27.8 | 1.4 |
| | 40 | 30.9 | 1.0 | 28.8 | 0.8 | 17.0 | 1.8 | 29.7 | 0.9 | 22.3 | 1.1 | 31.0 | 0.8 | 38.1 | 1.6 | 45.6 | 1.4 |
| −10 | 4 | 22.3 | 1.8 | 22.6 | 1.9 | 18.5 | 0.8 | 23.1 | 1.4 | 17.1 | 3.5 | 25.2 | 0.7 | 12.7 | 1.6 | 17.0 | 1.2 |
| | 20 | 44.9 | 0.9 | 55.7 | 1.1 | 56.9 | 1.2 | 57.0 | 0.3 | 36.2 | 0.7 | 39.7 | 1.3 | 46.2 | 1.4 | 56.3 | 0.6 |
| | 28 | 45.2 | 0.4 | 56.6 | 0.4 | 52.5 | 0.2 | 68.5 | 0.4 | 32.4 | 2.2 | 51.2 | 0.3 | 44.2 | 2.6 | 64.7 | 0.3 |
| | 40 | 39.1 | 1.6 | 47.1 | 0.8 | 42.5 | 0.4 | 52.7 | 0.3 | 26.8 | 2.0 | 35.4 | 1.0 | 28.3 | 1.0 | 36.2 | 1.4 |
| −31.6 | 4 | 19.1 | 2.4 | 21.1 | 0.4 | 16.6 | 3.3 | 32.1 | 2.7 | 13.8 | 0.8 | 17.8 | 0.5 | 26.2 | 2.0 | 34.3 | 0.7 |
| | 20 | 34.0 | 1.2 | 45.5 | 2.2 | 40.8 | 1.8 | 48.4 | 1.4 | 29.2 | 2.5 | 35.8 | 4.6 | 69.2 | 0.9 | 71.3 | 0.8 |
| | 28 | 50.5 | 1.2 | 57.4 | 1.5 | 29.2 | 1.1 | 35.6 | 1.3 | 36.9 | 4.8 | 43.5 | 0.5 | 59.5 | 1.0 | 61.8 | 0.5 |
| | 40 | 36.3 | 1.8 | 59.8 | 2.0 | 50.7 | 1.3 | 57.5 | 0.7 | 20.3 | 0.9 | 25.3 | 0.7 | 47.4 | 1.8 | 50.7 | 2.4 |

### 3.3. Relative Proportion of $NO_3^-$ to Total Mineral N

We observed four patterns for the relative proportion of nitrate to total mineral N depending on the GM type, temperature, and humidity (Figure 3). GM1 was affected by a decrease in humidity (i.e., a suction decrease), with a weak influence of temperature. GM2 was affected mostly at 40 °C and in the driest and wettest conditions (−3.2 and −31.6 kPa, respectively). GM3 was affected by temperature but not by humidity; the ratio was almost the same for all temperatures. The very low values of the ratio revealed that $NH_4^+$ accumulated substantially in this growing medium type, especially at 4 °C. GM4 was the least affected growing medium, with a slow but linear decrease in the values of the ratio as humidity decreased. These decreases were constant, but more pronounced at 20 and 28 °C than at 40 and 4 °C.

### 3.4. Temperature and Humidity Action Laws

A temperature action law was established for all growing media, all humidity levels, and by combining fertilizers F1 and F2 (Figure 4A). The model (Equation (1)) fitted the observed data well. It was calibrated for all humidity levels taken together. Table 4 presents the calibrated parameters and statistical performances (RMSE, $R^2$). The lowest RMSE and the best $R^2$ corresponded to the model adjustment with humidity at −10 kPa (Figure 4B).

The humidity action law f(H) is presented in Figure 5, combining fertilizers F1 and F2. Different patterns were observed depending on the growing medium, and they also changed according to temperature. At 28 °C, f(H) presented less variation for all growing media, with values around 1 in most cases (Figure 5B). This was almost the same at 40 °C, except in GM4 (Figure 5C). However, strong variations of f(H) were observed at 4 °C, except in GM1 where it was around 1 whatever the H-to-Hcc ratio (Figure 5A).

No correlation was found between the amount of mineralized N and the humidity level whatever the growing medium, but a relationship was established between the ratio of $NO_3^-$-N to total mineral N and the humidity content H (Figure 6). The ratio first increased with increasing H, whatever the temperature and considering all growing media, with a breakpoint of the slope when H reached 0.46 *v/v*, and a plateau thereafter. The segmented regression gave a very good correlation coefficient ($R^2 = 0.83$, $p < 0.001$).

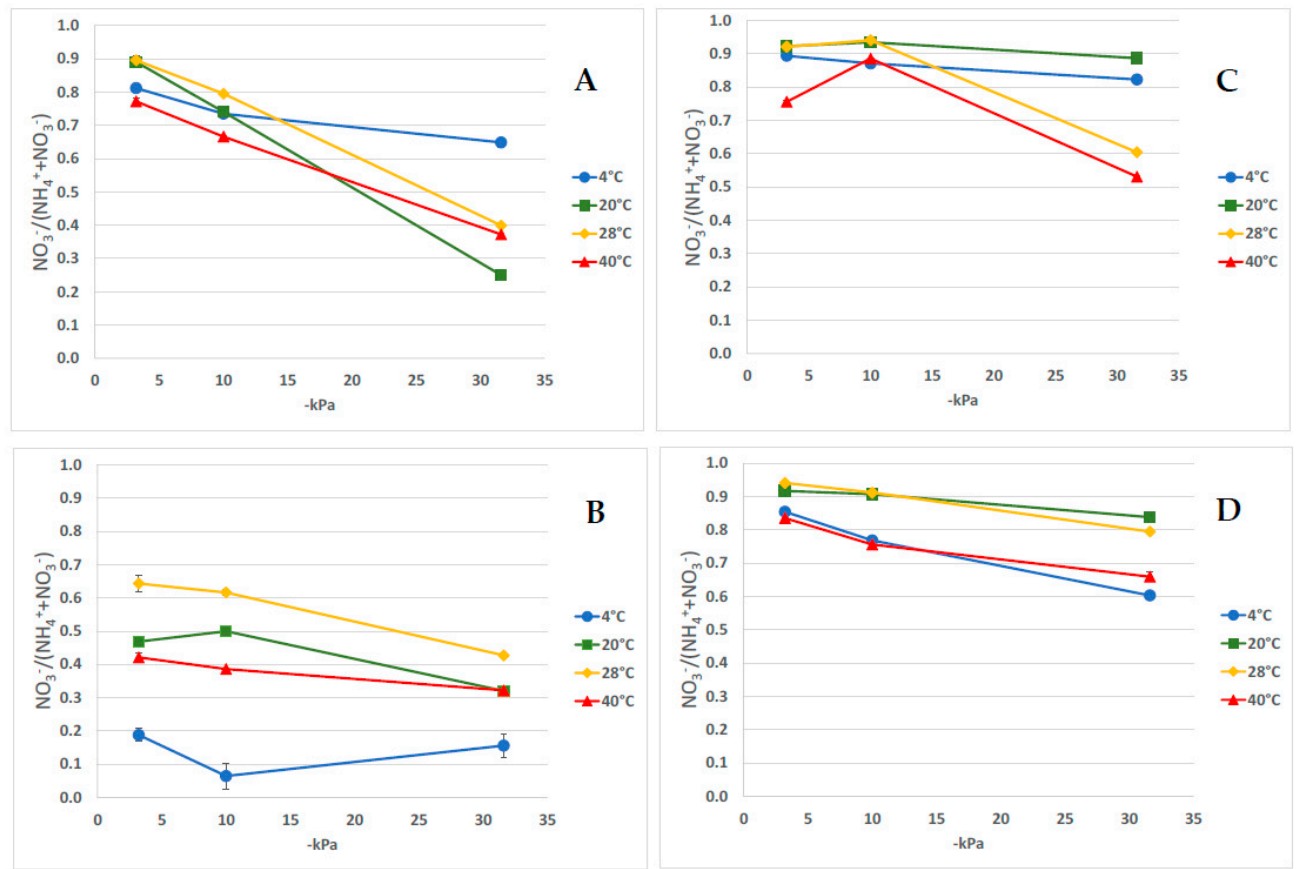

**Figure 3.** Relative proportion of nitrate to total mineral N (ratio: $NO_3^-$/total min N) depending on humidity and temperature in GM1 (**A**), GM2 (**B**), GM3 (**C**), and GM4 (**D**) fertilized with F2. Bars represent standard deviations (*n* = 3).

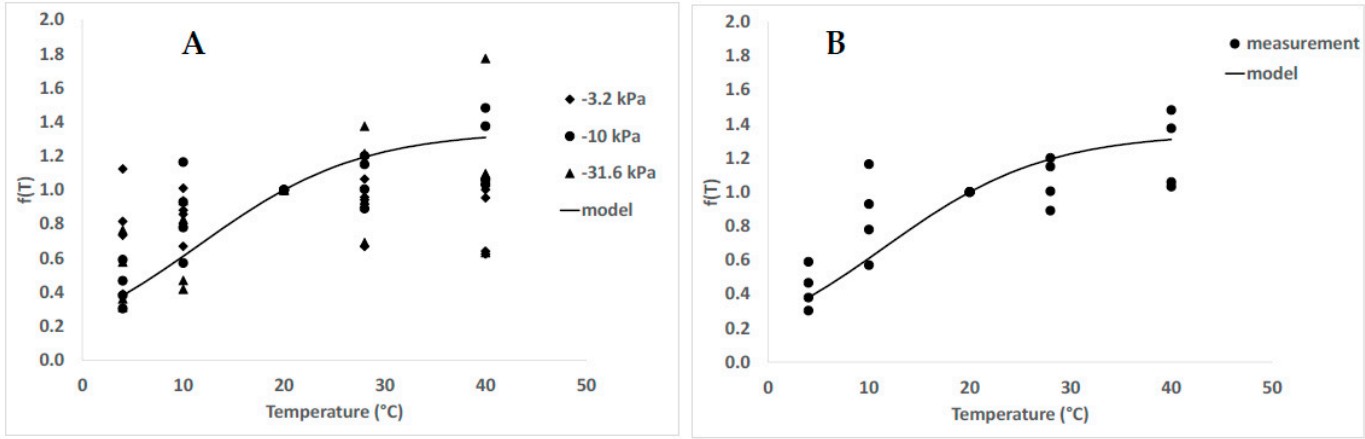

**Figure 4.** (**A**) Temperature response during N mineralization for all growing media incubated at −3.2, −10, and −31.6 kPa; (**B**) temperature response of all growing media incubated at −10 kPa. Data are the means of six replicates (i.e., three each with fertilizers F1 and F2). The temperature action law f(T) was calculated according to Equation (1).

**Table 4.** Adjusted parameters B and k and statistical performance of the temperature action law model.

|  | −3.2 kPa | −10 kPa | −31.6 kPa | All Suction Treatments |
|---|---|---|---|---|
| B | 1.00 | 1.20 | 1.16 | 1.35 |
| k | 0.25 | 0.12 | 0.12 | 0.12 |
| RMSE | 0.21 | 0.14 | 0.26 | 0.24 |
| $R^2$ | 0.30 ns | 0.91 *** | 0.70 *** | 0.66 *** |

*** $p < 0.001$, ns: nonsignificant.

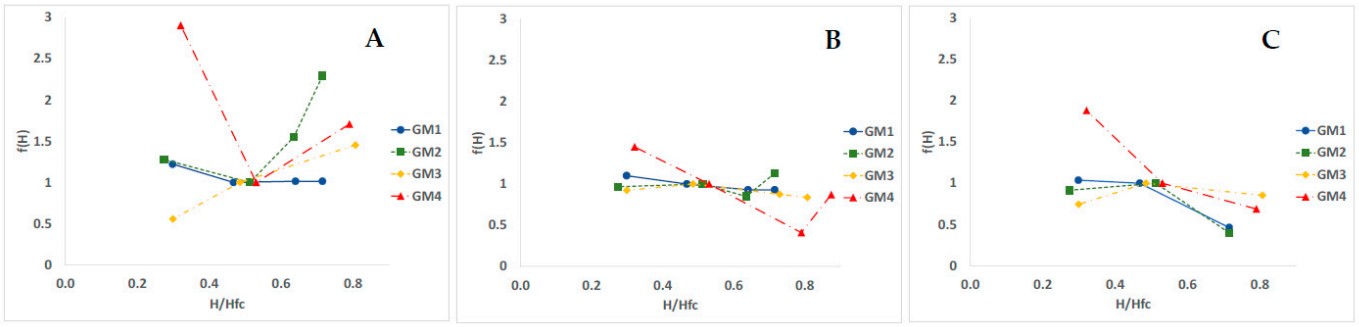

**Figure 5.** Water content action law at (**A**) 4 °C, (**B**) 28 °C, and (**C**) 40 °C for the four growing media. Data are the means of six replicates.

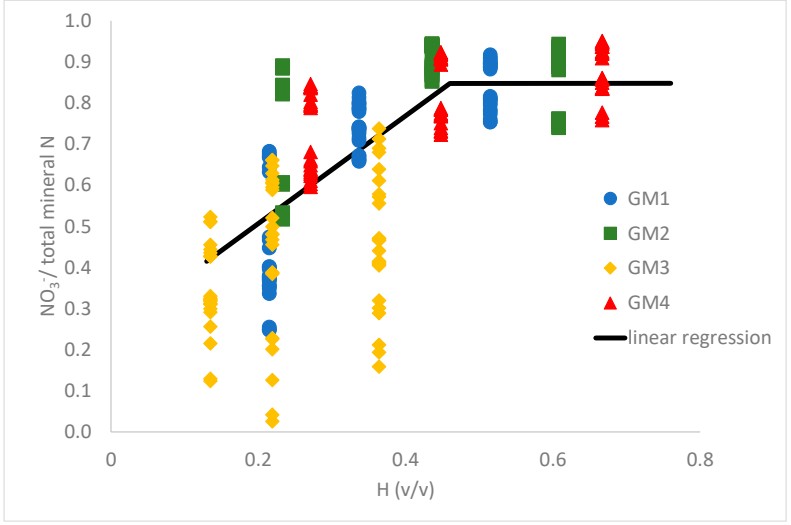

**Figure 6.** Effect of growing media volumetric water content on the average ratio of $NO_3^-$ to total mineral N. The dataset compiles all temperatures.

## 4. Discussion

In soils, it has been widely demonstrated that N mineralization is particularly dependent on temperature [35], humidity [36], and texture [37,38]. Growing media are made of organically stable compounds that strongly limit biological activity in the absence of fertilization [10]. Compared to soils, growing media do not provide available nutrients, especially nitrogen, and they usually require mineral or organic fertilization for biological activation of microbes and plant growth. Studies on growing media are mostly focused on physical properties such as hydrodynamic parameters and aeration, or they lay the emphasis on chemical properties such as water pH, electrical conductivity, or cationic exchange capacity [39–42]. These properties are very important and easily monitored, but they weakly reflect the biological aspects and nutrient availability needed for organically fertilized crops. Growers attempt to maintain these properties steadily throughout cropping, but alteration of growing media compounds and root growth can modify them [43,44].

This study was focused on the monitoring of the N mineralization process—which depends on many bio-physicochemical interactions—in four commercial organic growing media with similar physicochemical properties (Table 1). We tested different temperature and moisture levels and compared the responses of the four growing media types when added with two different organic fertilizers and when no fertilizer was added.

The four growing media were relatively similar in terms of physicochemical properties even if they were made from different materials and for different cropping purposes. We rather focused the discussion on the biochemical and microbial aspects addressed in the body of the manuscript.

### 4.1. Growing Media Type

The organic growing media were biochemically stable and characterized by high C-to-nutrient ratios, much higher than the C:N:P stoichiometry of microbial biomass [45], which ranges between 42:6:1 and 60:7:1 according to [46] and [47], respectively. Due to their homeostatic stoichiometry, microbes are extremely constrained by the low resource availability in the growing medium; the growing media are considered as biologically inactive, such that microbes strongly respond to organic fertilization (Figure 1) [10]. Consequently, N immobilization can occur in growing media with a C:N ratio exceeding 30:1 due to microbial decomposition of available C—a process requiring N [48]. In the present study, the C:N ratios were similar: around 30 for three growing media types (GM1, GM2, and GM4), and twice higher in GM3 (Table 1). Thus, high N immobilization (i.e., a greater microbial demand) was expected in the four growing media, with the strongest effects in GM3. However, the time course of N mineralization did not confirm these expectations whatever the growing media or the temperature and humidity conditions when fertilizer was added. Even so, GM3 presented the best performances in terms of mineral N release (Figure 1C). Net N immobilization (i.e., negative net N mineralization rates due to gross N organization higher than gross N mineralization) occurred and was only observed in the absence of fertilization and in all growing media; it was the highest in GM2 and GM4, not in GM3 as expected (Figure 1). Net N immobilization was only observed in growing media that already contained mineral N and plant-based compost (i.e., GM2 and GM4). Consequently, its intensity appeared to be limited by the very low initial mineral N content of GM3. Significant N mineralization occurred at 40 °C in the absence of fertilization. However, we failed to distinguish whether it resulted from the activation of microbes mineralizing growing media compounds or from dead cells, since this temperature can be critical for some microbial populations such as nitrifiers [49] (Supplementary Materials Table S6). These results could be confirmed by decreased N mineralization rates at this temperature in the fertilized treatments, but further investigations are required. N immobilization occurred in all four commercial growing media, especially because they received a compost fraction during formulation that produced mineral nitrogen before they were used. However, this immobilization effect was easily outperformed by organic fertilization. On the contrary, the raw materials showed no such immobilization effect and even no biological activity or very weak activity in the absence of organic fertilization [10].

### 4.2. Fertilizer Type

F1 and F2 were commercial animal-based and plant-based fertilizers, respectively. Due to confidentiality rules, the fertilizer compositions were unavailable, but we analyzed them for pH, elemental composition, and C:N:P stoichiometry (Table 2). We detected significant differences between the two organic fertilizers (Table S1). However, the patterns obtained with either fertilizer were very close, and we only showed curves of unfertilized versus F2-fertilized growing media to clarify illustrations (Figures 1 and 2). We previously showed that the huge C:N and C:P ratios of different growing media were the most important drivers of organic fertilizer mineralization and microbial activity, and they constrained the autochthonous microbial communities through C and nutrient availability [10]. The results of the current study confirm these effects. The C:N ratios of the growing media

indeed largely overcame (Table 1) the different C:N ratios of the two fertilizers, and this explained the small difference in their N mineralization response. The C:N ratio of an organic fertilizer is usually considered as a good predictor of the N mineralization or immobilization balance following fertilizer incorporation in soil [50,51]. However, in growing media where nutrient availability is low, fertilizers have a smaller impact than growing media on N mineralization, and this raises the question of the importance of growing media formulation. Granular fertilizers present similar patterns of nutrient release, but other organic fertilizers such as raw or thermally treated horn can have a huge impact on the control of nitrogen release, while remaining driven by the unbalanced stoichiometry of the growing media types [10]. Since C cannot be decoupled from the N and P cycles [10], we hypothesized that the lower P content in F2 (Table 2) may also have increased the microbial mining effect [52] to access P compared to F1, leading to overall faster fertilizer mineralization. Further investigations on coupling microbial C, N, and P functions [10] would be necessary to confirm this assumption.

We also expressed total mineral N at the end of incubation as a percentage of N fertilizer addition and detailed the patterns according to temperature and humidity, as a function of the two fertilizers (F1 and F2, Table 3). However, these values were not cumulative because we only measured the mineral N content at different timepoints of incubation. Thus, at the end of the experiment, the mineral N content did not express the total mineral N produced from the organic fertilizers, but the total mineral N content as a fraction of fertilizer-added N. Only a minimum value of what was really mineralized from the organic fertilizers was expressed. We cannot rule out that some of the mineral N came from (i) microbial turnover [53], especially at 40 °C, which can be critical for some microbial populations, and (ii) growing media biodegradation. However, growing media biodegradation is believed to be very low in growing media and would necessitate specific C inputs to trigger a priming effect [54].

*4.3. Temperature Effects and Action Law*

Temperature influences transformation rates through the responses of microorganisms and enzymatic activities. We tested four temperatures frequently met during plant growth in horticulture. Specifically, 20 and 28 °C are classical temperatures in the greenhouse and supposed to be optimal, whereas 4 and 40 °C are extreme temperatures affecting nutrient availability by slowing down microbial activity; 40 °C potentially affects the microbes themselves. We observed maximum nitrification at 28 °C, close to the soil optimum of 30 °C [55]. However, 40 °C sometimes gave the best mineral N content depending on the growing media, suggesting that this temperature provided for the highest mineralization rates. Delving deeper into the ammonification and nitrification processes (Figure S2), this mineralization was not sustainable since the $NH_4^+$ content was higher (Figure S1) than at the other temperatures while the $NO_3^-$ content decreased, indicating that nitrifiers were probably affected [55,56]. This unbalance between ammonification and nitrification was also analyzed by studying the relative proportion of $NO_3^-$ content over total mineral N (Figure 3): 40 °C and mostly 4 °C consistently resulted in bad conditions, and even critical ones for GM3.

N mineralization increases exponentially within the range of temperatures met in farmed soils (0–40 °C) and can be successfully modeled with numerous functions [57]. This study showed that the formalism of the temperature action law proposed by the STICS model for soils is also adapted to organic growing media mineralization. This is a first modeling of the effect of temperature on N mineralization rates applied to growing media. Modeling performance was best when using incubations were run at −10 kPa, as this modality was most adequate to reveal fertilizer N mineralization. A common way to express temperature sensitivity is to use the Q10 function. A Q10 of 2, for example, means that the rate of a particular process doubles when the temperature increases by 10 °C [58]. Using 20 °C as the reference temperature, the Q10 value of the growing media

was 1.13, lower than the average Q10 value of arable soils, but within the large range of values reported in the literature (from 0.55 to 11.9 [59]).

### 4.4. Humidity Effect and Action Law

Matric suction ($\psi$) was used to study the effect of humidity (H). However, due to their composition, the growing media had a specific H at a given $\psi$ value, and this made it more difficult to analyze the results (Table 5). H might have been a better parameter choice in the experimental design, even though $\psi$ affected organic nitrogen mineralization statistically. The choice of $\psi$ made agronomic sense in terms of water-filled porosity; a $\psi$ of $-1$ kPa is equivalent to the retention capacity of a growing media (H$fc$), a $\psi$ of $-10$ kPa corresponds to the temporary wilting point, and a $\psi$ of $-100$ kPa corresponds to the permanent wilting point [60].

**Table 5.** Volumetric water content ($v/v$) and water-filled pore space (WFPS, %) values according to the growing media and $\psi$ modalities.

|  | GM1 | GM2 | GM3 | GM4 |
|---|---|---|---|---|
| $\theta$ at 0 kPa ($v/v$) | 0.891 | 0.883 | 0.920 | 0.880 |
| $\theta$ at $-3.2$ kPa ($v/v$) | 0.515 | 0.608 | 0.364 | 0.667 |
| $\theta$ at $-10$ kPa ($v/v$) | 0.337 | 0.436 | 0.219 | 0.448 |
| $\theta$ at $-31.6$ kPa ($v/v$) | 0.215 | 0.233 | 0.135 | 0.271 |
| WFPS $-3.2$ kPa (%) | 57.8 | 68.9 | 39.6 | 75.8 |
| WFPS $-10$ kPa (%) | 37.8 | 49.4 | 23.8 | 50.9 |
| WFPS $-31.6$ kPa (%) | 24.1 | 26.4 | 14.7 | 30.8 |

Water is necessary for microbial activity, and its content has to be balanced with the oxygen required for root and microbial respiration [61]. Aerobic microbial activity is optimal at a humidity volumetric content ranging between 50% and 70% of the water holding capacity (WHC) [62,63] corresponding to water and oxygen availability in good equilibrium. Other studies estimate the maximum microbial activity (respiration and nitrification) in soils around 60% of the total porosity occupied by water (WFPS, "water-filled pore space") [64,65]. A major influence of the water content has been shown on microorganism activity in different organic growing media, with higher microbial respiration at 63% WFPS compared to 73% and 83% WFPS [66]. According to the theory mentioned above, fertilizer mineralization should be optimal around $-3.2$ kPa, as in GM1 and GM3. Yet, the mineralization rates of GM2 and GM4 were highest at $-31.6$ kPa, i.e., the driest conditions of this study. As a result, oxygen could be more important than water availability. Suction of 31.6 kPa is supposed to be too low for plant survival and growth. The ratio of $NO_3^-$ to ($NH_4^+ + NO_3^-$) as a function of growing media humidity showed a similar trend to that observed for soils, regardless of temperature [67], i.e., a progressive increase with humidity up to 0.46 $v/v$ (corresponding to a WFPS of 50–52% depending on the growing medium), followed by a plateau up to 0.67 $v/v$ (corresponding to a WFPS of 73–76% depending on the growing media) (Figure 6). Thus, the optimal humidity for nitrification in the growing media was similar to that of soils. However, we failed to establish a humidity action law on the basis of experimental data. Indeed, at a given temperature each growing media had a specific response curve to H/H$fc$ (Figure 5); when H/H$fc$ increased, f(H) decreased for some growing media and then increased with increasing H/H$fc$ values. For other growing media, f(H) progressively increased, or even reached a plateau. At another temperature, the growing media behaved again in a different way, making it difficult to establish a generic humidity action law.

Thus, an interaction between growing media temperature and moisture did occur, and this process is commonly encountered in soils [68,69]. Several humidity action laws have been established for arable soils and expressed as a function of the soil water content, the WFPS, or the matric potential. The need to improve the representation of this relationship in models has been highlighted. The authors of [70] presented a data-driven analysis of soil humidity–respiration relations based on 90 soils. They showed how the relationship

between soil heterotrophic respiration and different soil humidity levels is consistently affected by soil properties. On the basis of the proportional response of soil respiration (PRSR) related to a 0.01 increase in soil humidity as the central unit for analysis, they found little or no effect of soil properties on the PRSR in organic soils (i.e., a soil organic content higher than 300 g·kg$^{-1}$). Thus, due to their nature and their specific behavior, the formalisms known today are not adapted to growing media; research work has to be developed in this area.

*4.5. Management Implications*

In terms of professional applications, growing media and fertilizer type need to be considered at the same time to determine the rate of N release adequate for plant growth as precisely as possible. Depending on plant requirements, professionals could select a growing media according to its use (but physical characteristics would need to be checked) and use an organic fertilizer to provide nutrients. Our results tend to show that GM4 supplied slow nutrient release that quickly reached a plateau at the lowest level in this study, whereas GM3 supplied slowly the highest nutrient level. These results also indicate that detecting the plateau would have required a longer incubation time than 49 days for GM3 (Figure 1). Moreover, all four growing media presented a linear increase in total mineral N at 40 °C indicating that extreme temperature can cause fast N release with potential N loss if not synchronized with plant needs, whereas the process would be best controlled at a temperature maintained between 20 and 28 °C. Peat is a reference material in soilless production. Nevertheless, its use is questioned because exploiting peatland implies depleting a recognized carbon (C) sink. Thus, efforts and the immediate need for peat reduction in horticulture are a strong challenge for the future. Growing media tested in this study were peat-based, but combined with other alternative materials. We were able to show here that growing media with only 50% or 60% peat (GM2 and GM4, respectively) mineralized as much fertilizer as GM1 (80% of peat). The results are, therefore, encouraging and demonstrate that it is possible to progressively free ourselves from peat.

Mineralized F2 induced a high N release and appeared to be interesting for short- or medium-term crop cultivation, while the fertilizer dose could be defined accordingly. The temperature and $\psi$ conditions that promoted the highest N fertilizer mineralization were 28 °C and −10 kPa (for GM2 and GM3) or −31.6 kPa (for GM1 and GM4). To go further, growing media–fertilizer combinations should be tested in actual growth experiments since the plant uptake could reveal inadequate growing media–fertilizer combinations or too limiting ones in terms of nutrient supply, as suspected with GM4. Indeed, synchronizing nutrient supply with the nutrient requirements of plants is a major issue for increasing nutrient use efficiency. While nitrogen is essential and often the most limiting element for plant growth, plants can be subjected to multiple nutrient limitations, especially colimitation of N and P [71]. A depressive effect of organic versus mineral fertilization is frequently observed [72,73]. For example, higher ammonification over nitrification rates is the main explanation for the lower performances of organically grown basil plant because roots are exposed to high levels of $NH_4^+$ without supplying enough $NO_3^-$ [74,75].

**5. Conclusions**

The N mineralization dynamics of two organic fertilizers in four growing media types at different temperature and humidity conditions showed a strong impact of the different treatments on $NH_4^+$ and $NO_3^-$ release. Under optimal conditions of temperature (20 °C) and humidity (−10 kPa), 32% to 57% of the applied fertilizer was mineralized after 49 days depending on the growing media. These results constitute major food for thought on fertilizer application strategies during crop itineraries. The introduction of plants in the system will have an impact on the mineralization process, which we plan to study in the future. We attempted to adapt temperature and humidity action laws, whose formalisms are derived from work on soils, to growing media. We succeeded in describing the effect of temperature with an action law common to the four growing media, but the response of

the growing media to humidity greatly varied among growing media and in a temperature-dependent manner. Therefore, we failed to establish an action law for humidity, although a satisfactory relationship between nitrification and humidity was demonstrated. Research is needed to further investigate the effect of humidity and temperature-humidity interactions on the mineralization of organic N from fertilizers. In addition, the present results need to be refined using other growing media–fertilizer pairs. This work will allow for the short-term development of a prediction model of mineralization of organic N from fertilizers in soilless growing media production because such a model is lacking at present.

**Supplementary Materials:** The following are available online at https://www.mdpi.com/article/10.3390/horticulturae8020152/s1: Figure S1. Influence of temperature on $NH_4^+$-N at $-10$ kPa water matric suction, for (A) GM1, (B) GM2, (C) GM3, and (D) GM4; Figure S2. Influence of temperature on $NO_3^-$-N content at $-10$ kPa water matric suction, for (A) GM1, (B) GM2, (C) GM3, and (D) GM4; Figure S3. Influence of water matric suction on $NH_4^+$-N content at 20 °C, for (A) GM1, (B) GM2, (C) GM3, and (D) GM4; Figure S4. Influence of water matric suction on $NO_3^-$-N content at 20 °C, for (A) GM1, (B) GM2, (C) GM3, and (D) GM4; Figure S5. $NO_3^-$ to total N min ratio, depending on matric suction and temperature and with fertilizer F1, in (A) GM1, (B) GM2, (C) GM3, and (D) GM4. Bars represent standard deviation ($n = 3$); Table S1. Results of three-way repeated-measures ANOVA with growing media, fertilizer (Fert), temperature (Temp), and matric water suction ($\psi$) as between subject and time (t) after fertilizer addition as within subject, on total mineralized N, $NH_4^+$, and $NO_3^-$ contents; Table S2. $NO_3^-$, $NH_4^+$, and total mineral N in GM1, depending on $\psi$, temperature, and fertilizers modalities ($n = 3$, F0 = without fertilizer); Table S3. $NO_3^-$, $NH_4^+$, and total mineral N in GM2, depending on $\psi$, temperature, and fertilizer modalities ($n = 3$, F0 = without fertilizer); Table S4. $NO_3^-$, $NH_4^+$, and total mineral N in GM3, depending on $\psi$, temperature, and fertilizer modalities ($n = 3$, F0 = without fertilizer); Table S5. $NO_3^-$, $NH_4^+$, and total mineral N in GM4, depending on $\psi$, temperature, and fertilizer modalities ($n = 3$, F0 = without fertilizer); Table S6. AmoA content (log nb_seq·g$^{-1}$ dw growing media) at $-3.2$ and $-31.6$ kPa, and temperatures of 20 and 40 °C, during the 49 day incubation ($n = 3$).

**Author Contributions:** Conceptualization, S.B., M.V., P.C., R.G. and M.B.; methodology, S.B., M.V., P.C., R.G. and M.B.; validation, S.B., M.V., P.C., R.G., M.B. and S.R.; formal analysis, P.C., R.G., S.R., M.V. and M.B., investigation, P.C., R.G., S.R., M.V., S.B. and M.B.; writing—original draft preparation, P.C., R.G. and S.R.; writing—review and editing, P.C., R.G., S.R., M.V., L.P., S.B. and M.B.; visualization, P.C., R.G., S.R., M.V., L.P., S.B. and M.B.; supervision, P.C., R.G. and S.R.; project administration, S.B.; funding acquisition, S.B., M.V., P.C., R.G., M.B. and S.R. All authors have read and agreed to the published version of the manuscript.

**Funding:** This research was funded by CASDAR, OptiFaz project, grant number 5746.

**Institutional Review Board Statement:** Not applicable.

**Informed Consent Statement:** Not applicable.

**Data Availability Statement:** Data sharing not applicable.

**Conflicts of Interest:** The authors declare no conflict of interest. The funders had no role in the design of the study; in the collection, analyses, or interpretation of data; in the writing of the manuscript, or in the decision to publish the results.

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
