# Peer review of "Organic Fertilization of Growing Media: Response of N Mineralization to Temperature and Moisture"

_horticulturae, doi:10.3390/horticulturae8020152_

Round 1
Reviewer 1 Report
The nutrient requirement of crops plays a key role in sustaining soil fertility and crop productivity those through a valuable balancing of organic fertilizers such as the possible uses of crop residues, manure, and compost . Under low-input agriculture systems where nutrient availability is a serious constraint for agriculture and food production, this practice is particularly important and determine cycle of C, N, and fertility of soil. Manure can serve as a source of important plant nutrients including phosphorus (P) and nitrogen (N). The main declared topic of the present study was the evaluation of time course of N mineralization by considering different fertilizer growing media pairs, in the absence of plants with in mind to consider abiotic factors as the main topic of mineralization cycle of N. That is a simplistic view of a very complex modelling factors that may affect the investigated cycle of N. It is well known that the interaction of environmental and soil factors by the composition or quality of the substrate are implemented in the transformations of N in soil, including indigenous soil N and applied N, moreover their values are strongly controlled with the microbiological factors but probably the Author has sterilized their growing medium before to start the experiment, I did not found any procedure about this process in the material and method section. Several paper has been previously published in order to determine the quantification of the effects of temperature (Griffins, T. S., C. W. Honeycutt, and Z. He. 2001. Effects of temperature, soil water status, and soil type on swine slurry nitrogen transformations. Biology and Fertility of Soils 36:442–446), soil water status (Drury, K. F., T. Q. Zhang, and B. D. Kay. 2003. The non-limiting and least limiting water ranges for soil nitrogen mineralization. Soil Science Society America Journal 67:1388–1404.), and soil texture (Sørensen, P., and E. S. Jensen. 1995. Mineralization of carbon and nitrogen from fresh and anaerobically stored sheep manure in soils with different texture and water contents. Biology and Fertility of Soils 19:29–35.; Thomsen, I. K., P. Schoenning, and B. T. Christensen. 2003. Mineralization of 15N-labelled sheep manure in soils of different texture and water contents. Biology and Fertility of Soils 37:295– 301) on many aspects of N availability, including the rate and extent of mineralization and nitrification, soil N retention, and potential environmental loss of N. Therefore the Author have to change the approach to the experimental data FIG 1 and FIG 2 show negative concentration levels that is unusual and have to be explain more clearly in the text. In FIG 1, 2, 3, 5 and 6 they use linear regression in order to interpolate data setting that did not show any linearity regression. The paper it is not publishable in the present form. The experimental design, the material and method and also data settings may be a valuable staring point to achieve conclusions that have to be confirm by an accurate multivariate analyses and modelling that are absent in the manuscript.
Author Response
Reviewer 1. Modifications are highlighted in yellow color in the revised manuscript
The nutrient requirement of crops plays a key role in sustaining soil fertility and crop productivity those through a valuable balancing of organic fertilizers such as the possible uses of crop residues, manure, and compost . Under low-input agriculture systems where nutrient availability is a serious constraint for agriculture and food production, this practice is particularly important and determine cycle of C, N, and fertility of soil. Manure can serve as a source of important plant nutrients including phosphorus (P) and nitrogen (N). The main declared topic of the present study was the evaluation of time course of N mineralization by considering different fertilizer growing media pairs, in the absence of plants with in mind to consider abiotic factors as the main topic of mineralization cycle of N. That is a simplistic view of a very complex modelling factors that may affect the investigated cycle of N. It is well known that the interaction of environmental and soil factors by the composition or quality of the substrate are implemented in the transformations of N in soil, including indigenous soil N and applied N, moreover their values are strongly controlled with the microbiological factors but probably the Author has sterilized their growing medium before to start the experiment, I did not found any procedure about this process in the material and method section
Answer: We completely agree with the reviewer's remark. Nitrogen mineralization results from a complex combination of biotic and abiotic factors. For this reason, the results of nitrogen mineralization presented here constitute a net mineralization, i.e. the completeness of all the factors governing this biological process. We thus manipulated the supposed main drivers of microbial activity (temperature and humidity, the conditions for life) and is well known in soil science studies. We are not in soil but in containers with organic growing media with a very weak microbial activity that need to be boosted with organic fertilizers. Growing media were not sterilized because they host specific microbial communities (see Montagne et al. 2015, 2017; Paillat et al. 2020 cited in the manuscript lines 48-59 and below), which are essential for the transformation of organic fertilizers into mineral elements.
Several paper has been previously published in order to determine the quantification of the effects of temperature (Griffins, T. S., C. W. Honeycutt, and Z. He. 2001. Effects of temperature, soil water status, and soil type on swine slurry nitrogen transformations. Biology and Fertility of Soils 36:442–446), soil water status (Drury, K. F., T. Q. Zhang, and B. D. Kay. 2003. The non-limiting and least limiting water ranges for soil nitrogen mineralization. Soil Science Society America Journal 67:1388–1404.), and soil texture (Sørensen, P., and E. S. Jensen. 1995. Mineralization of carbon and nitrogen from fresh and anaerobically stored sheep manure in soils with different texture and water contents. Biology and Fertility of Soils 19:29–35.; Thomsen, I. K., P. Schoenning, and B. T. Christensen. 2003. Mineralization of 15N-labelled sheep manure in soils of different texture and water contents. Biology and Fertility of Soils 37:295– 301) on many aspects of N availability, including the rate and extent of mineralization and nitrification, soil N retention, and potential environmental loss of N.
Answer: We are aware that a lot of knowledge on the factors influencing mineralization has been developed for more than 40 years in soils. In this manuscript, we made the assumption that the knowledge from soils could be transposed to growing media. The originality of this article is therefore to present original results of nitrogen mineralization in growing media. The introduction has been built in this sense, based on the current knowledge in this field in growing media. The references that you mention seem useful to us to present in the discussion to make the parallel between our growing media and soils Lines 426-427
Therefore the Author have to change the approach to the experimental data FIG 1 and FIG 2 show negative concentration levels that is unusual and have to be explain more clearly in the text.
Answer: the modalities for which negative values were observed correspond to some modalities without fertilizer. First mineral nitrogen was initially present in small quantities in all GM (Table 1). In addition, GM2, GM3 and GM4 contained compost that was allowed to mineralize. Thus, the modalities without fertilizer for which negative values were observed are the result of a net N immobilization by the microorganisms. Such explanations were already given L462-468
In FIG 1, 2, 3, 5 and 6 they use linear regression in order to interpolate data setting that did not show any linearity regression.
Answer: we are not sure if we understood the reviewer's remark. Linearity regressions model are not presented in the present manuscript, except in fig6, where linear regressions were tested (L214-216) and were significant (L393-394). If the reviewer's remark concerns the linearity of the points between each measurement date, in order not to bias interpretations of the temporal dynamics of a process, it is conventional to connect the points by linear lines. This is in fact the case in the articles you quoted (see Sørensen et al., 1995, and Drury et al., 1993). Especially since for nitrogen these are not "instantaneous" data but cumulative quantities over time, and therefore the dates are linked together, even if we do not know exactly what happened between two points.
The experimental design, the material and method and also data settings may be a valuable starting point to achieve conclusions that have to be confirm by an accurate multivariate analyses and modelling that are absent in the manuscript.
Answer: We used 3-way repeated measures ANOVA (rmANOVA) to test the interaction of the growing medium type (GM), the fertilizer type (Fert), temperature (Temp) and matric water suction (ψ) on NO3-, NH4+ and NH4++NO3- concentrations following fertilization (L207-209). This analysis is in supplementary materials (table S1). Moreover, we have made a PCA analysis but the information did not appear more than the mass of data that we already present in detail
We have not presented the modeling results because taken experimental + modelling together, this will have made for an overly dense article, with methodological developments that are too long and diverse. We precised in the manuscript L631-634 that perspectives will focuse on modelling. A second manuscript will present the results of two approaches (1) a multivariate statistical model and (2) a first order kinetic model as developed for N mineralization in soils.
Reviewer 2 Report
The present study is examining the responses of N mineralization of Temperature and moisture on growing media.
The present study is of interest and well structured. The results and discussion sections are described appropriately in order to yield the mentioned conclusions.
To my opinion the present work can be accepted
- L38. In soilless culture actually, we do not have limitations in nutrients. It seems authors wanted to stress other points, however, as it is written here, it is not correct.
- Since the graphs are busy, it would help to have colored figures, since the journal is accepting that (if the authors have already colored figs.)
Some suggestions/comments
- L32, L36. Use a common way for soilless
- Check the reference list.
- Journal names are in the full title and in abbreviations
- Publication years is not always in bold
- Full stop or not at the end of each reference
Author Response
The present study is examining the responses of N mineralization of Temperature and moisture on growing media. The present study is of interest and well structured. The results and discussion sections are described appropriately in order to yield the mentioned conclusions. To my opinion the present work can be accepted
Answer: thank you for your positive comments
L38. In soilless culture actually, we do not have limitations in nutrients. It seems authors wanted to stress other points, however, as it is written here, it is not correct.
Answer: we agree with this comment, we left “nutrients” from the sentence
Since the graphs are busy, it would help to have colored figures, since the journal is accepting that (if the authors have already colored figs.)
Answer: we agree with reviewer proposition, figures are now in colour
L32, L36. Use a common way for soilless
Answer: this was corrected
Check the reference list. Journal names are in the full title and in abbreviations. Publication years is not always in bold. Full stop or not at the end of each reference
Answer: we have made all the corrections except for the year which should be in bold for journals only
Reviewer 3 Report
This paper is very interesting and can be accepted after minor revisions. The most important changes suggested are:
L.32-33. The subject is thorny in horticulture, particularly in soilless production which consumes resources (water and other inputs). This statement makes no sense. Soil-grown production also consumes water and other inputs. Generally, in the introduction the authors are advised to avoid populistic approaches about organic – inorganic fertilization and objectively state the real advantages of organic fertilization: 1. It introduces a recycling concept in agroecosystems. 2. No use of synthetic inorganic N fertilizers reduces considerably the CO2 emissions produced during the industrial N2 fixation.
L. 132 The standard SI unit to express moisture tension (or suction) is Paskal and its multiples (kPa, MPa). The authors are advised to replace pF values with kPa throughout the ms.
Author Response
Reviewer 3. Modifications are highlighted in blue color in the revised manuscript
This paper is very interesting and can be accepted after minor revisions. The most important changes suggested are:
L.32-33. The subject is thorny in horticulture, particularly in soilless production which consumes resources (water and other inputs). This statement makes no sense. Soil-grown production also consumes water and other inputs. Generally, in the introduction the authors are advised to avoid populistic approaches about organic – inorganic fertilization and objectively state the real advantages of organic fertilization: 1. It introduces a recycling concept in agroecosystems. 2. No use of synthetic inorganic N fertilizers reduces considerably the CO2 emissions produced during the industrial N2 fixation.
Answer: Thanks for your comment which is very relevant, we have improved the introduction Lines 35-38
L132 The standard SI unit to express moisture tension (or suction) is Pascal and its multiples (kPa, MPa). The authors are advised to replace pF values with kPa throughout the ms.
Answer: we followed reviewer’s recommendation throughout the manuscript